# A Neural Local Coherence Model

## Abstract

We propose a local coherence model based on a convolutional neural network that operates over the entity grid representation of a text. The model captures long range entity transitions along with entity-specific features without loosing generalization, thanks to the power of distributed representation. We present a pairwise ranking method to train the model in an end-to-end fashion on a task and learn task-specific high level features. Our evaluation on three different coherence assessment tasks demonstrates that our model achieves state of the art results outperforming existing models by a good margin.

## 1 Introduction and Motivation

What distinguishes a coherent text from a random sequence of sentences is that it binds the sentences together to express a meaning as a whole — the interpretation of a sentence usually depends on the meaning of its neighbors. *Coherence models* that can distinguish a coherent from incoherent texts have a wide range of applications in text generation, summarization, and coherence scoring.

Several formal theories of coherence have been proposed (Mann and Thompson, 1988a; Grosz et al., 1995; Asher and Lascarides, 2003), and their principles have inspired development of existing coherence models (Barzilay and Lapata, 2008; Lin et al., 2011; Li and Hovy, 2014). Among these models, the *entity grid* (Barzilay and Lapata, 2008), which is based on Centering Theory (Grosz et al., 1995), is arguably the most popular, and has seen a number of improvements over the years. As shown in Figure 2, the entity grid model represents a text by a grid that captures how grammatical roles of different entities change from

sentence to sentence. The grid is then converted into a feature vector containing probabilities of local entity transitions, which enables machine learning models to learn the degree of text coherence. Extensions of this basic grid model incorporate entity-specific features (Elsner and Charniak, 2011), multiple ranks (Feng and Hirst, 2012), and coherence relations (Feng et al., 2014).

While the entity grid and its extensions have been successful in many applications, they are limited in several ways. Firstly, they use discrete representation for grammatical roles and features, which limits the model to consider sufficiently long transitions (Bengio et al., 2003). Secondly, feature vector computation in existing models is decoupled from the target task, which limits the models to learn task-specific features.

In this paper, we propose a *neural* architecture for coherence assessment that can capture long range entity transitions along with arbitrary entity-specific features. Our model obtains generalization through *distributed representations* of entity transitions and entity features. We also present an end-to-end training method to learn task-specific high level features automatically in our model.

We evaluate our approach on three different evaluation tasks: discrimination, insertion, and summary coherence rating, proposed previously for evaluating coherence models (Barzilay and Lapata, 2008; Elsner and Charniak, 2011). Discrimination and insertion involve identifying the right order of the sentences in a text with different levels of difficulty. In summary coherence rating task, we compare the rankings, given by the model, against human pairwise judgments of coherence.

The experimental results show that our neural models consistently improve over the non-neural counterparts (i.e., existing entity grid models) yielding absolute gains of about $4\%$ on discrimination, up to $2.5\%$ on insertion, and more

than $4\%$ on summary coherence rating. Our model achieves state of the art results in all these tasks. We have released our code with the submission.

The remainder of this paper is organized as follows. We describe entity grid, its extensions, and its limitations in Section 2. In Section 3, we present our neural model. We describe evaluation tasks and results in Sections 4 and 5. We give a brief account of related work in Section 6. Finally, we conclude with future directions in Section 7.

## 2 Entity Grid and Its Extensions

Motivated by Centering Theory (Grosz et al., 1995), Barzilay and Lapata (2008) proposed an entity-based model for representing and assessing text coherence. Their model represents a text by a two-dimensional array called *entity grid* that captures transitions of discourse entities across sentences. As shown in Figure 2, the rows of the grid correspond to sentences, and the columns correspond to discourse entities appearing in the text. They consider noun phrases (NP) as entities, and employ a coreference resolver to detect mentions of the same entity (e.g., *Obama*, *the president*). Each entry $G_{i,j}$ in the entity grid represents the syntactic role that entity $e_j$ plays in sentence $s_i$, which can be one of: subject (**S**), object (**O**), or other (**X**). In addition, entities not appearing in a sentence are marked by a special symbol (**-**). If an entity appears more than once with different grammatical roles in the same sentence, the role with the highest rank (S $\succ$ O $\succ$ X) is considered.

To represent an entity grid with a feature vector, Barzilay and Lapata (2008) compute probability for each local entity transition of length $k$ (i.e., $\{S, O, X, -\}^k$), and represent each grid by a vector of $4^k$ transitions probabilities. To distinguish between transitions of important entities from unimportant ones, they consider the *salience* of the entities, which they quantify by their occurrence frequency in the document. Assessment of text coherence is then formulated as a ranking problem in an SVM preference ranking framework (Joachims, 2002).

Subsequent studies proposed to extend the basic entity grid model. Filippova and Strube (2007) attempted to improve the model by grouping entities based on semantic relatedness, but did not get significant improvement. Elsner and Charniak (2011) proposed a number of improvements. They initially show significant improvement by includ-

| | UNIT | PRODUCTS | RESEARCH | COMPANY | PARTS | CONTROLS | INDUSTRY | ELECTRONICS | TERM | CONCERN | AEROSPACE | EMPLOYEES | SERVICES | LOS ANGELES | EATON |
|---|---|---|---|---|---|---|---|---|---|---|---|---|---|---|---|
| $s_0$ | O | – | X | X | – | – | – | – | – | – | – | X | – | – | X |
| $s_1$ | – | – | – | – | – | – | – | – | S | – | – | – | – | – | – |
| $s_2$ | – | O | – | – | – | – | X | – | – | – | – | O | O | X | – |
| $s_3$ | – | – | – | – | X | X | – | X | – | O | X | – | – | – | S |

$s_0$: Eaton Corp. said it sold its Pacific Sierra Research unit to a company formed by employees of that unit.

$s_1$: Terms were not disclosed.

$s_2$: Pacific Sierra, based in Los Angeles, has 200 employees and supplies professional services and advanced products to industry.

$s_3$: Eaton is an automotive parts, controls and aerospace electronics concern.

Figure 1: Entity grid representation (top) for a document (below) from WSJ (id: 0079).

ing non-head nouns (i.e., nouns that do not head NPs) as entities in the grid.[1] Then, they extend the grid to distinguish between entities of different types by incorporating entity-specific features like named entity, noun class, modifiers, etc. These extensions led to the best results reported so far.

Entity grid and its extensions have been successfully applied to many downstream tasks including coherence rating (Barzilay and Lapata, 2008), essay scoring (Burstein et al., 2010), story generation (McIntyre and Lapata, 2010), and readability assessment (Pitler et al., 2010; Barzilay and Lapata, 2008). They have also been critical components in state-of-the-art sentence ordering models (Soricut and Marcu, 2006; Elsner and Charniak, 2011; Lin et al., 2011).

### 2.1 Limitations of Entity Grid Models

Despite its success, existing entity grid models are limited in several ways.

• Existing models use discrete representation for grammatical roles and features, which leads to the so-called **curse of dimensionality** problem (Bengio et al., 2003). In particular, to model transitions of length $k$ with $\mathcal{R}$ different grammatical roles, the basic entity grid model needs to compute $\mathcal{R}^k$ transition probabilities from a grid. One can imagine that the estimated distribution becomes sparse as $k$ increases. This limits the model to consider longer transitions – existing models use $k \leq 3$.

---

[1]They match the nouns to detect coreferent entities.

This problem is exacerbated when we want to include entity-specific features, as the number of parameters grows exponentially with the number of features (Elsner and Charniak, 2011).

• Existing models compute feature representations from entity grids in a task-agnostic way. In other words, feature extraction is decoupled from the target downstream tasks. This can limit the models to learn task-specific features. Therefore, models that can be trained in an end-to-end fashion on different target tasks are desirable.

In the following section, we present a neural architecture that allows us to capture long range entity transitions along with arbitrary entity-specific features without loosing generalization. We also present an end-to-end training method to learn task-specific features automatically.

## 3 The Neural Coherence Model

Figure 2 summarizes our neural architecture for modeling local coherence, and how it can be trained in a pairwise fashion. The architecture takes a document as input, and first extracts its entity grid.[2] The first layer of the neural network transforms each grammatical role in the grid into a distributed representation, a real-valued vector. The second layer computes high-level features by going over each column (transitions) of the grid. The following layer selects the most important high-level features, which are in turn used for coherence scoring. The features computed at different layers of the network are automatically trained by backpropagation to be relevant to the task. In the following, we elaborate on the layers of the neural network model.

**(I) Transforming grammatical roles into feature vectors:** Grammatical roles are fed to our model as indices taken from a finite vocabulary $\mathcal{V}$. In the simplest scenario, $\mathcal{V}$ contains $\{S, O, X, -\}$. However, we will see in Section 3.1 that as we include more entity-specific features, $\mathcal{V}$ can contain more symbols. The first layer of our network maps each of these indices into a distributed representation $\mathbb{R}^d$ by looking up a shared embedding matrix $E \in \mathbb{R}^{|\mathcal{V}| \times d}$. We consider $E$ a model parameter to be learned by backpropagation on a given task. We can initialize $E$ randomly or using pretrained vectors trained on a general coherence task.

---

[2]For clarification, pairwise input as shown in the figure is required only to train the model.

Given an entity grid $G$ with columns representing entity transitions over sentences in a document, the lookup layer extracts a $d$-dimensional vector for each entry $G_{i,j}$ from $E$. More formally,

$$\mathcal{L}(G) = \left\langle E(G_{1,1}) \; \cdots \; E(G_{i,j}) \; \cdots \; E(G_{m,n}) \right\rangle \tag{1}$$

where $E(G_{i,j})$ refers to the row in $E$ that corresponds to the grammatical role $G_{i,j} \in \mathcal{V}$; $m$ is the total number of sentences and $n$ is the total number of entities in the document. The output $\mathcal{L}(G)$ is a tensor in $\mathbb{R}^{m \times n \times d}$, which is fed to the next layer of the network as we describe below.

**(II) Modeling entity transitions:** The vectors produced by the lookup layer are combined by subsequent layers of the network to generate a coherence score for the document. To compose higher-level features from the embedding vectors, we make the following modeling assumptions:

• Similar to existing entity grid models, we assume there is no spatio-temporal relation between the entities in a document. In other words, columns in a grid are treated independently.

• We are interested in modeling entity transitions of arbitrary lengths in a *location-invariant* way. This means, we aim to compose local patches of entity transitions into higher-level representations, while treating the patches independently of their position in the entity grid.

Under these assumptions, the natural choice to tackle this problem is to use a *convolutional* approach, used previously to solve other NLP tasks (Collobert et al., 2011; Kim, 2014).

**Convolution layer:** A convolution operation involves applying a *filter* $\mathbf{w} \in \mathbb{R}^{k.d}$ (i.e., a vector of weight parameters) to each entity transition of length $k$ to produce a new abstract feature

$$h_t = f(\mathbf{w}^T \mathcal{L}_{t:t+k-1,j} + b_t) \tag{2}$$

where $\mathcal{L}_{t:t+k-1,j}$ denotes the concatenation of $k$ vectors in the lookup layer representing a transition of length $k$ for entity $e_j$ in the grid, $b_t$ is a bias term, and $f$ is a nonlinear activation function, e.g., ReLU (Nair and Hinton, 2010) in our model.

We apply this filter to each possible $k$-length transitions of different entities in the grid to generate a *feature map*, $\mathbf{h}^i = [h_1, \cdots, h_{m.n+k-1}]$. We

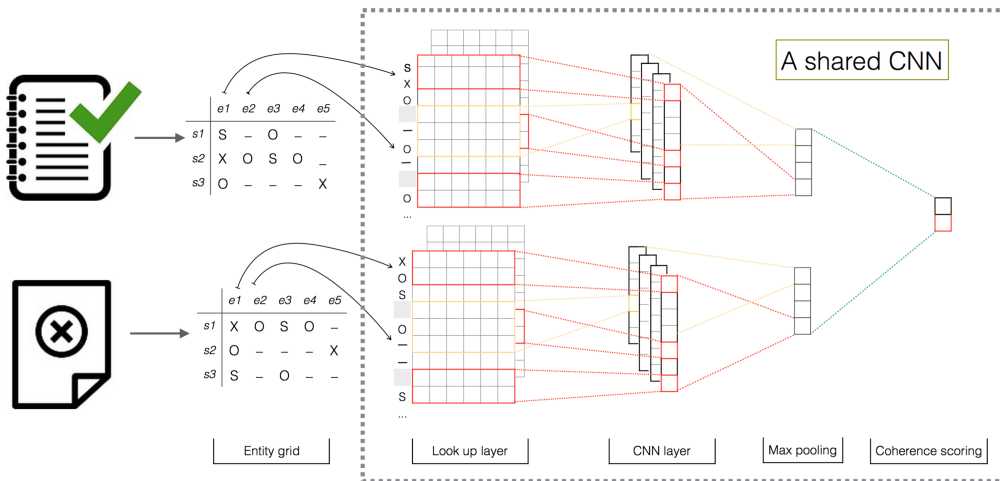

Figure 2: Neural architecture for modeling local coherence and the pairwise training method.

repeat this process $N$ times with $N$ different filters to get $N$ different feature maps (Figure 2). Notice that we use a *wide* convolution (Kalchbrenner et al., 2014), as opposed to *narrow*, to ensure that the filters reach entire columns of a grid, including the boundary entities. This is done by performing *zero-padding*, where out-of-range (i.e., for $t < 0$ or $t > \{m, n\}$) vectors are assumed to be zero.

Convolutional filters learn to compose local transition features of a grid into higher-level representations automatically. Since it operates over the distributed representation of grid entries, compared to traditional grid models, the transition length $k$ can be sufficiently large (e.g., $5 - 8$ in our experiments) to capture long-range transitional dependencies without overfitting on the training data. Moreover, unlike existing grid models that compute transition probabilities from a single document, embedding vectors and convolutional filters are learned from all training documents, which helps the neural framework to obtain better generalization and robustness.

**Pooling layer:** After the convolution, we apply a *max-pooling* operation to each feature map.

$$\mathbf{m} = [\mu_p(\mathbf{h}^1), \cdots, \mu_p(\mathbf{h}^N)] \qquad (3)$$

where $\mu_p(\mathbf{h}^i)$ refers to the $\max$ operation applied to each non-overlapping[3] window of $p$ features in the feature map $\mathbf{h}^i$. Max-pooling reduces the output dimensionality by a factor of $p$, and it drives the model to capture the most salient local features

---

[3] We set the *stride size* to be the same as the pooling length $p$ to get non-overlapping regions.

from each feature map in the convolutional layer.

**Coherence scoring:** Finally, the max-pooled features are used in the output layer of the network to produce a coherence score $y \in \mathbb{R}$.

$$y = \mathbf{v}^T \mathbf{m} + b \qquad (4)$$

where $\mathbf{v}$ is the weight vector and $b$ is a bias term.

**Why it works:** Intuitively, each filter detects a specific transition pattern (e.g., 'SS-O-X' for a coherent text), and if this pattern occurs somewhere in the grid, the resulting feature map will have a large value for that particular region and small values for other regions. By applying max pooling on this feature map, the network then discovers that the transition appeared in the grid.

### 3.1 Incorporating Entity-Specific Features

Our model as described above neuralizes the basic entity grid model that considers only entity transitions without distinguishing between types of the entities. However, as Elsner and Charniak (2011) pointed out entity-specific features could be crucial for modeling local coherence. One simple way to incorporate entity-specific features into our model is to attach the feature value (e.g., named entity type) with the grammatical role in the grid. For example, if an entity $e_j$ of type PERSON appears as a subject (S) in sentence $s_i$, the grid entry $G_{i,j}$ can be encoded as PERSON-S.

### 3.2 Training

Our neural model assigns a coherence score to an input document $d$ based on the degree of lo-

cal coherence observed in its entity grid $G$. Let $y = \phi(G|\theta)$ define our model that transforms an input grid $G$ to a coherence score $y$ through a sequence of lookup, convolutional, pooling, and linear projection layers with parameter set $\theta$. The parameter set $\theta$ includes the embedding matrix $E$, the filter matrix $W$, the weight vector $\mathbf{v}$, and the biases. We use a *pairwise ranking* approach (Collobert et al., 2011) to learn $\theta$.

The training set comprises *ordered* pairs $(d_i, d_j)$, where document $d_i$ exhibits a higher degree of coherence than document $d_j$. As we will see in Section 4 such orderings can be obtained automatically or through manual annotation. In training, we seek to find $\theta$ that assigns a higher coherence score to $d_i$ than to $d_j$. We minimize the following ranking objective with respect to $\theta$:

$$\mathcal{J}(\theta) = \max\{0, 1 - \phi(G_i|\theta) + \phi(G_j|\theta)\} \quad (5)$$

where $G_i$ and $G_j$ are the entity grids corresponding to documents $d_i$ and $d_j$, respectively. Notice that (also shown in Figure 2) the network shares its layers (and hence $\theta$) to obtain $\phi(G_i|\theta)$ and $\phi(G_j|\theta)$ from a pair of input grids $(G_i, G_j)$.

Barzilay and Lapata (2008) adopted a similar ranking criterion using an SVM preference kernel learner as they argue coherence assessment is best seen as a ranking problem as opposed to classification (*coherent* vs. *incoherent*). Also, the ranker gives a scoring function $\phi$ that a text generation system can use to compare alternative hypotheses.

## 4 Evaluation Tasks

We evaluate the effectiveness of our coherence models on two different evaluation tasks: sentence ordering and summary coherence rating.

### 4.1 Sentence Ordering

Following (Elsner and Charniak, 2011), we evaluate our models on two sentence ordering tasks: discrimination and insertion.

In the *discrimination* task (Barzilay and Lapata, 2008), a document is compared to a random permutation of its sentences, and the model is considered correct if it scores the original document higher than the permuted one. We use 20 permutations of each document in the test set in accordance with previous work.

In the *insertion* task (Elsner and Charniak, 2011), we evaluate models based on their ability

| | Sections | # Doc. | # Pairs | Avg. # Sen. |
|---|---|---|---|---|
| TRAIN | 00-13 | 1,378 | 26,422 | 21.5 |
| TEST | 14-24 | 1,053 | 20,411 | 22.3 |

Table 1: Statistics on WSJ dataset.

to locate the original position of a sentence previously removed from a document. To measure this, each sentence in the document is removed in turn, and an insertion place is located for which the model gives the highest coherence score to the document. The insertion score is then computed as the average fraction of sentences per document reinserted in their actual position.

Discrimination can be easier for longer documents, since a random permutation is likely to be different than the original one. Insertion is a much more difficult task since the candidate documents differ only by the position of one sentence.

**Dataset:** For sentence ordering tasks, we use the Wall Street Journal (WSJ) portion of Penn Treebank, as used by (Elsner and Charniak, 2008, 2011; Lin et al., 2011; Feng et al., 2014). Table 1 gives basic statistics about the dataset. Following previous works, we use 20 random permutations of each article, and we exclude permutations that match the original document.[4] The fourth column (# Pairs) in Table 1 shows the resulting number of (*original*, *permuted*) pairs used for training our model and for testing in the discrimination task.

Some previous studies (Barzilay and Lapata, 2008; Li and Hovy, 2014) used the AIRPLANES and the EARTHQUAKES corpora, which contain reports on airplane crashes and earthquakes, respectively. Each of these corpora contains 100 articles for training and 100 articles for testing. The average number of sentences per article in these two corpora is 10.4 and 11.5, respectively.

We preferred WSJ corpus for several reasons. First and most importantly, WSJ corpus is larger than other corpora (see Table 1). Large training set is crucial for learning effective deep learning models (Collobert et al., 2011), and a large enough test set is necessary to make a general comment about model performance. Secondly, as Elsner and Charniak (2011) pointed out, texts in AIRPLANES and EARTHQUAKES are constrained in style, whereas WSJ documents are more like normal informative articles. Thirdly, we could re-

---

[4]Short articles may produce many matches.

produce results on this dataset for the competing systems (e.g., entity grid and its extensions) using publicly available Brown coherence toolkit.[5]

## 4.2 Summary Coherence Rating

We further evaluate our models on the summary coherence rating task proposed by Barzilay and Lapata (2008), where we compare rankings given by a model to a pair of summaries against rankings elicited from human judges.

**Dataset:** The summary dataset was extracted from the Document Understanding Conference (DUC'03), which contains 6 clusters of multi-document summaries produced by human experts and 5 automatic summarization systems. Each cluster has 16 summaries of a document with pairwise coherence rankings given by humans judges; see (Barzilay and Lapata, 2008) for details on the annotation method. There are 144 pairs of summaries for training and 80 pairs for testing.

## 5 Experiments

In this section, we present our experiments — the models we compare, their settings, and the results.

### 5.1 Models Compared

We compare our coherence model against a random baseline and several existing models.

**Random:** The Random baseline makes a random decision for the evaluation tasks.

**Graph-based Model:** This is the graph-based unsupervised model proposed by Guinaudeau and Strube (2013). We use the implementation from the cohere[6] toolkit (Smith et al., 2016), and run it on the test set with syntactic projection (command line option 'projection=3') for graph construction. This setting yielded best scores for this model.

**Grid-all nouns (E&C):** This is the simple extension of the original entity grid model, where all nouns are considered as entities. Elsner and Charniak (2011) report significant gains by considering all nouns as opposed to only head-nouns. Results for this model were obtained by training the baseline entity grid model (command line option '-n') in Brown coherence toolkit on our dataset.

**Extended grid (E&C):** This represents the extended entity grid model of Elsner and Charniak (2011) that uses 9 entity-specific features; 4 of them were computed from external corpora. This model considers all nouns as entities. For this system, we train the extended grid model (command line option '-f') in Brown coherence toolkit.

**Grid-CNN:** This is our proposed neural extension of the basic entity grid (all nouns), where we only consider entity transitions as input.

**Extended Grid-CNN:** This corresponds to our neural model that incorporates entity-specific features following the method described in Section 3.1. To keep the model simple, we include only three entity-specific features from (Elsner and Charniak, 2011) that are easy to compute and do not require any external corpus. The features are: (*i*) *named entity* type, (*ii*) *salience* as determined by occurrence frequency of the entity, and (*iii*) whether the entity has a *proper* mention.

**A remark:** We also experimented with the distributed sentence model proposed recently by Li and Hovy (2014). We trained it on our WSJ corpus using their code[7] with the same setting that produced the best results on the discrimination task. However, the results on our dataset were very disappointing – accuracy of only $19.34\%$ in discrimination. We were not sure what could go wrong, therefore, we excluded it from our table of results.

### 5.2 Settings for Neural Models

We held out $10\%$ of the training documents to form a development set (DEV) on which we tune the hyper-parameters of our neural models. For discrimination and insertion tasks, the resulting DEV set contains 138 articles and 2,678 pairs after removing the permutations that match the original documents. For the summary rating task, DEV contains 14 pairs of summaries.

We implement our models in Theano (Theano Development Team, 2016). We use rectified linear units (ReLU) as activations ($f$). The embedding matrix is initialized with samples from uniform distribution $\mathcal{U}(-0.01, 0.01)$, and the weight matrices are initialized with samples from glorot-uniform distribution (Glorot and Bengio, 2010).

We train the models by optimizing the pairwise ranking loss in Equation 5 using the gradient-based online learning algorithm RMSprop with

---

[5]https://bitbucket.org/melsner/browncoherence
[6]https://github.com/karins/CoherenceFramework

[7]http://cs.stanford.edu/ bdlijiwei/code/

|  | Batch | Emb. | Dropout | Filter | Win. | Pool |
|---|---|---|---|---|---|---|
| Grid-CNN | 128 | 100 | 0.5 | 150 | 6 | 6 |
| Ext. Grid-CNN | 32 | 100 | 0.5 | 150 | 5 | 6 |

Table 2: Optimal hyper-parameter setting for our neural models based on development set accuracy.

|  | Discr. | | Ins. |
|---|---|---|---|
|  | Acc | $F_1$ | |
| Random | 50.0 | 50.0 | 12.60 |
| Graph-based (G&S) | 64.23 | 65.01 | 11.93 |
| Grid-all nouns (E&C) | 81.58 | 81.60 | 22.13 |
| Extended Grid (E&C) | 84.95 | 84.95 | 23.28 |
| Grid-CNN | 85.57† | 85.57† | 23.12 |
| Extended Grid-CNN | **88.69†** | **88.69†** | **25.95†** |

Table 3: Results on **Discr**imination and **Ins**ertion tasks. † indicates a neural model is significantly superior to its non-neural counterpart with p-value < 0.01.

parameters ($\rho$ and $\epsilon$) set to the values suggested by Tieleman and Hinton (2012).[8] We use up to 25 epochs. To avoid overfitting, we use dropout (Srivastava et al., 2014) of hidden units, and do *early stopping* by observing accuracy on the DEV set – if the accuracy does not increase for 10 consecutive epochs, we exit with the best model recorded so far. We search for optimal mini**batch** size in $\{16, 32, 64, 128\}$, **emb**edding size in $\{80, 100, 200\}$, **dropout** rate in $\{0.2, 0.3, 0.5\}$, **filter** number in $\{100, 150, 200, 300\}$, **win**dow size in $\{2, 3, 4, 5, 6, 7, 8\}$, and **pool**ing length in $\{3, 4, 5, 6, 7\}$. Table 2 shows the optimal hyper-parameter setting for our models. The best model on DEV is then used for the final evaluation on the TEST set. We run each experiment five times, each time with a different random seed, and we report the average of the runs to avoid any randomness in results. Statistical significance tests are done using an *approximate randomization* test based on the accuracy. We used SIGF V.2 (Padó, 2006) with 10,000 iterations.

### 5.3 Results on Sentence Ordering

Table 3 shows the results on discrimination and insertion tasks. Among the existing models, the graph-based model gets the lowest scores, where the extended grid gets the highest scores on both tasks. By neuralizing the basic grid model (Grid-

---
[8]Other adaptive algorithms, e.g., ADAM (Kingma and Ba, 2014), ADADELTA (Zeiler, 2012) gave similar results.

|  | Acc | $F_1$ |
|---|---|---|
| Random | 50.0 | 50.0 |
| Graph-based (G&S) | 80.0 | 81.5 |
| Grid (B&L) | 83.8 | - |
| Grid-CNN | 85.0 | 85.0 |
| Extended Grid-CNN | 86.3 | 86.3 |
| Pre-trained Grid-CNN | 86.3 | 86.3 |
| Pre-trained Ext. Grid-CNN | **87.5** | **87.5** |

Table 4: Results on **Summary Coherence Rating** tasks.

all nouns), our Grid-CNN model delivers absolute improvements of about $4\%$ in discrimination and $1\%$ in insertion. When we compare our Extended Grid-CNN with its non-neural counterpart Extended Grid, we observe similar gains in discrimination and more gains ($2.5\%$) in insertion. Note that the Extended Grid-CNN yields these improvements considering only a subset of the Extended Grid features. This demonstrates the effectiveness of distributed representation and convolutional feature learning method.

Compared to discrimination, gain in insertion is less verbose. There could be two reasons. First, as mentioned before, insertion is a harder task than discrimination. Second, our models were not trained specifically on the insertion task. The model that is trained to distinguish an original document from its random permutation may learn features that are not specific enough to distinguish documents when only one sentence differs. It will be interesting to see how the model performs when we train it on the insertion task directly.

### 5.4 Results on Summary Coherence Rating

Table 4 presents the results on the summary coherence rating task, where we compare our models with the graph-based method and the reported results of Barzilay and Lapata (2008) on the same experimental setting.[9] Since there are not many training instances, our neural models may not learn well for this task. Therefore, we also present versions of our model, where we use pre-trained models from discrimination task on WSJ corpus (last two rows in the table ). The pre-trained models are then *fine-tuned* on the summary rating task.

We can observe that even without pre-training

---
[9]The extended grid model does not use pairwise training, therefore could not be trained on the summarization dataset.

our models outperform existing models, and pre-training gives further improvements. Specifically, Pre-trained Grid-CNN gives an improvement of 2.5% over the Grid model, and including entity features pushes the improvement further to 3.7%.

## 6 Related Work

Barzilay and Lapata (2005, 2008) introduced the entity grid representation of discourse to model local coherence that captures the distribution of discourse entities across sentences in a text. They also introduced three tasks to evaluate the performance of coherence models: discrimination, summary coherence rating, and readability.

A number of extensions of the basic entity grid model has been proposed. Elsner and Charniak (2011) included entity-specific features to distinguish between entities. Feng and Hirst (2012) used the basic grid representation, but improved its learning to rank scheme. Their model learns not only from original document and its permutations but also from ranking preferences among the permutations themselves. Guinaudeau and Strube (2013) convert a standard entity grid into a bipartite graph representing entity occurrences in sentences. To model local entity transition, the method constructs a directed projection graph representing the connection between adjacent sentences. Two sentences have a connected edge if they share at least one entity in common. The coherence score of the document is then computed as the average out-degree of sentence nodes.

In addition, there are some approaches that model text coherence based on coreferences and discourse relations. Elsner and Charniak (2008) proposed the discourse-new model by taking into account mentions of all referring expression (i.e., NPs) whether they are first mention (*discourse-new*) or subsequent (*discourse-old*) mentions. Given a document, they run a maximum-entropy classifier to detect each NP as a label $L_{np} \in \{new, old\}$. The coherence score of the document is then estimated by $\prod_{np:NPs} P(L_{np}|np)$. In this work, they also estimate text coherence through pronoun coreference modeling. Lin et al. (2011) assume that a coherent text has certain discourse relation patterns. Instead of modeling entity transitions, they model discourse role transitions between sentences. In a follow up work, Feng et al. (2014) trained the same model but using features derived from deep discourse structures annotated with Rhetorical Structure Theory or RST (Mann and Thompson, 1988b) relations. Louis and Nenkova (2012) introduced a coherence model based on syntactic patterns in text by assuming that sentences in a coherent discourse should share the same structural syntactic patterns.

In recent years, there has been a growing interest in neuralizing traditional NLP approaches – language modeling (Bengio et al., 2003), sequence tagging (Collobert et al., 2011), syntactic parsing (Socher et al., 2013), and discourse parsing (Li et al., 2014), etc. Following this tradition, in this paper we propose to neuralize the popular entity grid models. Li and Hovy (2014) also proposed a neural framework to compute coherence score of a document by estimating coherence probability for every window of $L$ sentences (in their experiments, $L = 3$). First, they use a recurrent or a recursive neural network to compute the representation for each sentence in $L$ from its words and their pre-trained embeddings. Then the concatenated vector is passed through a non-linear hidden layer, and finally the output layer decides if the window of sentences is a coherent text or not. Our approach is fundamentally different from their approach; our model operates over entity grids, and we use convolutional architecture to model sufficiently long entity transitions.

## 7 Conclusion and Future Work

We presented a local coherence model based on a convolutional neural network that operates over the distributed representation of entity transitions in the grid representation of a text. Our architecture can model sufficiently long entity transitions, and can incorporate entity-specific features without loosing generalization power. We described a pairwise ranking approach to train the model on a target task and learn task-specific features. Our evaluation on discrimination, insertion and summary coherence rating tasks demonstrates the effectiveness of our approach yielding the best results reported so far on these tasks.

In future, we would like to include other sources of information in our model. Our initial plan is to include rhetorical relations, which has been shown to benefit existing grid models (Feng et al., 2014). We would also like to extend our model to other forms of discourse, especially, asynchronous conversations, where participants communicate with each other at different times (e.g., forum, email).

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

## A   Supplemental Material

