# Peer review of "A Neural Local Coherence Model"

_ACL 2017 — decision unknown_

[Official Review · Reviewer 1 · rating 4 · confidence 3]
soundness 5 · originality 5 · clarity 4 · impact 3 · substance 4 · appropriateness 5 · meaningful comparison 3 · presentation format Poster

The paper introduces an extension of the entity grid model. A convolutional
neural network is used to learn sequences of entity transitions indicating
coherence, permitting better generalisation over longer sequences of entities
than the direct estimates of transition probabilities in the original model.

This is a nice and well-written paper. Instead of proposing a fully neural
approach, the authors build on existing work and just use a neural network to
overcome specific issues in one step. This is a valid approach, but it would be
useful to expand the comparison to the existing neural coherence model of Li
and Hovy. The authors admit being surprised by the very low score the Li and
Hovy model achieves on their task. This makes the reader wonder if there was an
error in the experimental setup, if the other model's low performance is
corpus-dependent and, if so, what results the model proposed in this paper
would achieve on a corpus or task where the other model is more successful. A
deeper investigation of these factors would strengthen the argument
considerably.

In general the paper is very fluent and readable, but in many places definite
articles are missing (e.g. on lines 92, 132, 174, 488, 490, 547, 674, 764 and
probably more). I would suggest proofreading the paper specifically with
article usage in mind. The expression "...limits the model to do X...", which
is used repeatedly, sounds a bit unusual. Maybe "limits the model's capacity to
do X" or "stops the model from doing X" would be clearer.

--------------

Final recommendation adjusted to 4 after considering the author response. I
agree that objective difficulties running other people's software shouldn't be
held against the present authors. The efforts made to test the Li and Hovy
system, and the problems encountered in doing so, should be documented in the
paper. I would also suggest that the authors try to reproduce the results of Li
and Hovy on their original data sets as a sanity check (unless they have
already done so), just to see if that works for them.

[Official Review · Reviewer 2 · rating 3 · confidence 3]
soundness 5 · originality 5 · clarity 4 · impact 3 · substance 3 · appropriateness 5 · meaningful comparison 3 · presentation format Poster

The paper proposes a convolutional neural network approach to model the
coherence of texts. The model is based on the well-known entity grid
representation for coherence, but puts a CNN on top of it. 

The approach is well motivated and described, I especially appreciate the clear
discussion of the intuitions behind certain design decisions (e.g. why CNN and
the section titled 'Why it works').

There is an extensive evaluation on several tasks, which shows that the
proposed approach beats previous methods. It is however strange that one
previous result could not be reproduced: the results on Li/Hovy (2014) suggest
an implementation or modelling error that should be addressed.

Still, the model is a relatively simple 'neuralization' of the entity grid
model. I didn't understand why 100-dimensional vectors are necessary to
represent a four-dimensional grid entry (or a few more in the case of the
extended grid). How does this help? I can see that optimizing directly for
coherence ranking would help learn a better model, but the difference of
transition chains for up to k=3 sentences vs. k=6 might not make such a big
difference, especially since many WSJ articles may be very short.

The writing seemed a bit lengthy, the paper repeats certain parts in several
places, for example the introduction to entity grids. In particular, section 2
also presents related work, thus the first 2/3 of section 6 are a repetition
and should be deleted (or worked into section 2 where necessary). The rest of
section 6 should probably be added in section 2 under a subsection (then rename
section 2 as related work).

Overall this seems like a solid implementation of applying a neural network
model to entity-grid-based coherence. But considering the proposed
consolidation of the previous work, I would expect a bit more from a full
paper, such as innovations in the representations (other features?) or tasks.

minor points:

- this paper may benefit from proof-reading by a native speaker: there are
articles missing in many places, e.g. '_the_ WSJ corpus' (2x), '_the_ Brown ...
toolkit' (2x), etc.

- p.1 bottom left column: 'Figure 2' -> 'Figure 1'

- p.1 Firstly/Secondly -> First, Second

- p.1 'limits the model to' -> 'prevents the model from considering ...' ?

- Consider removing the 'standard' final paragraph in section 1, since it is
not necessary to follow such a short paper.